# N-of-1 Trials as a Decision Support Tool in Clinical Practice: A Protocol for a Systematic Literature Review and Narrative Synthesis

**DOI:** 10.3390/healthcare7040136

**Published:** 2019-11-06

**Authors:** Joyce Samuel, Travis Holder, Donald Molony

**Affiliations:** 1McGovern Medical School, The University of Texas Health Science Center at Houston, 6431 Fannin St, Houston, TX 77030, USA; donald.a.molony@uth.tmc.edu; 2Houston Academy of Medicine, The Texas Medical Center Library, 1133 John Freeman Blvd, Houston, TX 77030, USA; travis.holder@library.tmc.edu

**Keywords:** n-of-1 trial, single case experimental design, personalized trial

## Abstract

The n-of-1 trial can utilized in clinical practice as a decision support tool, which may improve patient outcomes by providing both the patient and the clinician with objective evidence to inform personalized treatment decisions. As its use broadens, it will be important to study whether the added time and effort of an n-of-1 trial results in measurable improvements in important patient outcomes compared to usual clinical practice. Parallel-group randomized clinical trials testing the n-of-1 approach versus usual care have been undertaken in a number of medical settings. A systematic review will be performed according to PRISMA guidelines, using MEDLINE, Embase, Cochrane, CINAHL, PsycINFO, Scopus, and Web of Science to search for randomized clinical trials in humans, without date or language restriction. Reports from the gray literature and ongoing studies in trial registries will be included. Articles will be screened by two independent reviewers with a third reviewer consulted to adjudicate disagreement. The quality of included studies will be assessed using the Cochrane Collaboration’s tool for assessing risk of bias. A narrative synthesis will explore the differing methodological approaches of the included studies. The protocol will be registered in the PROSPERO registry, and the results of the review will be published in a peer-reviewed journal. To our knowledge, this systematic review will be the first to comprehensively assess the existing research on randomized trials testing the n-of-1 trial approach in clinical practice.

## 1. Introduction

N-of-1 trials, or personalized trials, represent an innovative scientific strategy that can be viewed simultaneously as both research and clinical practice. One of the fundamental principles of evidence-based medicine is that clinical practice should begin with an understanding of the best available unbiased evidence. Answers to treatment questions can be garnered from parallel-group randomized clinical trials (RCTs), in which large groups of individuals are exposed to either the intervention in question or a control condition, and the average response of the groups is carefully measured and compared. However most clinicians are all too familiar with the limitations of this approach; the most important being (1) the dearth of RCTs undertaken in many of the conditions and populations that they are responsible for treating, and (2) the question of generalizability—that the individual’s response to a treatment may vary from the observed average treatment effect in an RCT. Given the remaining uncertainty inherent in clinical practice, n-of-1 trials have the potential to play an important role in patient care. In an n-of-1 trial, an individual patient is exposed to one or more interventions and a control condition in random order, and the individual’s response is carefully measured to estimate the relative treatment effects for the individual. N-of-1 trials can be used in clinical practice as a decision support tool by clinicians who must decide between two or more therapeutic alternatives for a chronic condition [1]. The ultimate goal of the n-of-1 trial as a clinical tool is to produce evidence to inform treatment plans that have been tailored to the individual based on his or her unique responses to the therapies under consideration. 

A number of case reports and case series describing the use of n-of-1 trials across a diverse spectrum of disease states and interventions have been published since their introduction to the medical landscape by Guyatt and colleagues in 1986 [2,3,4,5]. 

In usual clinical practice, prescribing decisions are often unsystematic and informed primarily by clinician preferences, anecdotal experience, and trial-and-error. Conversely, when used as a decision support tool, n-of-1 trials represent a treatment approach that reduces the guesswork inherent in usual practice. A variety of techniques can be employed to minimize bias, including randomized treatment order, objective outcomes assessment, and repeated treatment periods. Widespread use of n-of-1 trials in the clinical setting has been hindered by multiple factors, including a lack of awareness of the approach and skepticism about whether the n-of-1 trial results in measurable improvements in important patient outcomes compared to usual clinical practice, especially in light of the additional time and effort that may be required.

A systematic review of prospective randomized clinical trials is the gold standard in establishing whether any given intervention is superior to the alternative [6]. In this case, the intervention in question is the use of n-of-1 trials as a decision tool, and the alternative is usual clinical practice. The medical literature will be searched for RCTs testing groups of individuals whose treatment decisions were informed by either an n-of-1 trial or usual clinical practice.

The primary objective is to conduct a systematic review of the literature to determine whether n-of-1 trials have been shown in randomized trials to improve clinical outcomes compared to usual care. We will also critically appraise the methodological approaches of included studies to assess whether scientific rigor can be improved for future studies.

## 2. Materials and Methods 

This protocol was developed based on the Preferred Reporting Items for Systematic Reviews and Meta-Analysis for Protocols (PRISMA-P) 2015 statement, and will be registered in the PROSPERO international prospective registry [7]. Any important protocol amendments will be reported when publishing the results. The systematic review will be performed and reported according to the Preferred Reporting Items for Systematic Reviews and Meta-Analysis (PRISMA) guidelines [8]. 

### 2.1. Eligibility Criteria

We will search the medical literature for prospective, parallel-group, randomized clinical trials (RCTs) that compare a group of human subjects who were randomized to the n-of-1 trial strategy with another group who were randomized to another treatment approach and followed over time to assess a difference in any health outcome. We will restrict our search to medical n-of-1 trials. No restriction will be placed on language or date. 

The terminology used to designate n-of-1 of trials may be inconsistent across the body of literature on this topic. We will include only those RCTs testing n-of-1 trials that meet the following criteria: (1) randomized treatment periods within blocks or pairs, (2) crossover of interventions, (3) single patients as the unit of analysis for the n-of-1 trial. 

We will exclude case reports or case series of n-of-1 trials, n-of-1 observational designs, studies describing behavioral interventions, and expert reviews and commentaries.

### 2.2. Information Sources

The following databases will be electronically searched from inception: MEDLINE (PubMed), Embase, Cochrane Library, CINAHL, PsycINFO, Scopus, and Web of Science. We will search the gray literature using the Open Grey database and by consultation with experts. Unpublished or ongoing studies will be identified by searching ClinicalTrials.gov and the WHO international clinical trials registry platform. The reference lists of all included studies will be hand-searched. 

### 2.3. Search Strategy

A comprehensive search strategy is under development and will include (but not be limited to) “n-of-1”, “personalized trial”, “single case experimental design”, “multi-period crossover design”, and “single subject trial.” A second search will be conducted translating identified keywords, index terms, and necessary commands into each database.

### 2.4. Data Management and Study Selection

References will be managed in the screening and selection process using Rayyan, a web-based application [9]. After duplicates are removed, two investigators will screen all retrieved records by titles and subsequently by abstract, using the eligibility criteria described above. Disagreements over eligibility at the title and abstract level will be discussed first and persistent disagreements will be arbitrated by a third reviewer. For all articles with abstracts determined to be eligible, the full text will be retrieved and evaluated by two reviewers independently, with disagreements handled in the same manner as the title and abstract screening step. A flow diagram demonstrating the search and selection process will be developed according to PRISMA guidelines.

### 2.5. Data Collection

A standardized data extraction form will be developed and 2 investigators will independently extract the following data (see Table 1) regarding the overall RCT and the design of the individual n-of-1 trials separately.

### 2.6. Outcomes and Prioritization

All included studies will be summarized in a table of study characteristics, and we anticipate including the following main outcomes: primary outcome and whether comparison between groups was a summary of individual responses to treatment versus aggregated group response; acceptability of n-of-1 trial, including rates of drop-out and patient satisfaction; and cost analysis.

### 2.7. Risk of Bias in Individual Studies

Assessment of the completeness of the reporting will be measured using the Consolidated Standards of Reporting Trials (CONSORT) extension for n-of-1 trials (CENT) checklist [10]. For each RCT, the risk of bias will be assessed using the domain-based Cochrane Collaboration tool to evaluate risk of bias in randomized trials [11].

### 2.8. Data Synthesis

Quantitative meta-analysis will not be appropriate or possible due to anticipated differences in population, diagnoses, and interventions across included studies. We will use a narrative approach to provide summary descriptions of n-of-1 trial design, RCT study design, study quality, risk of bias, and outcomes reported.

### 2.9. Meta-Biases and Confidence in Cumulative Evidence

Risk of publication bias will be discussed, including why n-of-1 trials are particularly vulnerable due to inconsistent protocol registration. The Grading of Recommendations Assessment, Development and Evaluation (GRADE) criteria will be used to summarize the strength of the evidence to support the use of n-of-1 trials [12].

## 3. Discussion

To our knowledge, this review represents the first attempt to systematically review the body of evidence pertaining to the rigorous testing of n-of-1 trials against usual practice using RCTs. This systematic review will be submitted for publication in a peer-reviewed journal. The results will be used to identify whether more trials are needed, and may provide insight into how future RCTs may be designed with improved rigor to advance the science of n-of-1 trials.

## Figures and Tables

**Table 1 healthcare-07-00136-t001:** Data items to be extracted.

**Data Items about the Randomized Clinical Trial**	
	Study name, first author name, year of publication
	Country of the study
	Funding sources
	Institutional review board approval
	Clinical trial registry
	Trial setting (inpatient, outpatient, etc.)
	Disease and population under study
	Total number of subjects enrolled and sample size target
	Primary outcome Assessment of costs, harms, or patient satisfaction
	Randomization strategy
	Masking Subgroup analyses—whether any attempt was made to identify whether some patients benefited more from the n-of-1 strategy than others
**Data items about the n-of-1 trial being tested by the RCT**	
	Intervention type (pharmacological, surgical, behavioral)
	Number of treatments being compared
	Number of planned treatment crossovers
	Uniformity of n-of-1 trials across participants (interventions to be compared, number of treatment periods, length of treatment periods)
	Whether a paired randomization scheme was used (in which each patient receives an equal number of treatment “A” periods as treatment “B”)
	Treatment length and frequency
	Washout or minimization of carryover effects
	Blinding
	Primary outcome measurement tool and frequency Consideration of other treatment effects, including costs and harms Statistical methods, including whether carryover effects, period effects, and intrasubject correlation were considered
	Definition of responder or treatment success (statistical differences versus clinical significance)
	Description of heterogeneity of treatment effects, including Within-patient variability (treatment by patient interaction or proportion of patients with no treatment superiority) Between-patient variability (differences in the outcome for different patients or proportion of patients ultimately responding best to the same treatment)
	Number of individuals beginning and completing the n-of-1 trial
	Whether treatment was changed as a result of n-of-1 trial

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
