# Peer review of "N-of-1 Trials as a Decision Support Tool in Clinical Practice: A Protocol for a Systematic Literature Review and Narrative Synthesis"

_healthcare, 2019, doi:10.3390/healthcare7040136_

Round 1
Reviewer 1 Report
This is a very original project, and I am excited to see the results. This has potential across a variety of helping professions which have made use of an N=1 approach. I hope that you are able to find randomized trials of N=1 designs as decision tools! Introduction: I think it would be important to explain how the use of N=1 designs in practice facilitates decision making. I also think it is important to specify what constitutes an "N of 1" design. Do they need to be more rigorous designs, like "ABAB" or multiple-baseline designs, or is a simple AB sufficient? How many data points per phase? Or is this part of your CENT assessment? You might consider citing Barlow or Hayes who have been proponents of this approach for decades. Your methods for analysis need to be more explicit. Please avoid the "qualitative meta-analysis" term as it feels like nails on a chalk-board. You call it a narrative synthesis in the text, which is a better term. See Petticrew's work on narrative syntheses. In the abstract you mention using CONSORT (and the extension, which is appropriate) to assess the studies. However, CONSORT should not be used this way as it represents reporting guidelines and not risk of bias assessment. You mention the Cochrane RoB tool in your methods which is appropriate. Please edit your abstract for concurrence.As I said above, I think this is an exciting concept, and look forward to reading both the protocol and the completed review. Best Wishes!
Author Response
(reviewer comments in italics, author response in bold)
Thank you for your insightful comments and encouragement! All of your suggestions will be addressed and I believe the paper will be much stronger as a result. See my point by point responses below.
This is a very original project, and I am excited to see the results. This has potential across a variety of helping professions which have made use of an N=1 approach. I hope that you are able to find randomized trials of N=1 designs as decision tools!
Introduction: I think it would be important to explain how the use of N=1 designs in practice facilitates decision making.
Reviewer 2 raised similar concerns about the need to clarify that I am interested in evaluating n-of-1 as a treatment selection approach in the context of therapeutic uncertainty. I rewrote the title, abstract, and introduction to explicitly describe the use of n-of-1 designs as a decision tool in clinical care as opposed to purely an experimental/research approach.
I also think it is important to specify what constitutes an "N of 1" design. Do they need to be more rigorous designs, like "ABAB" or multiple-baseline designs, or is a simple AB sufficient? How many data points per phase? Or is this part of your CENT assessment? You might consider citing Barlow or Hayes who have been proponents of this approach for decades.
In anticipating not very many RCTs testing this approach, I plan to include even those with less rigorous designs but will specifically address these points with the risk of bias assessment for included studies. It may be important to include the less rigorous designs as I hypothesize that there is a difference in efficacy based on the methodological rigor of the n-of-1 design. In other words, the use of a less rigorous n-of-1 approach may not be any better for the patient compared to usual care; and I suspect that the more rigorous n-of-1 designs are more likely to result in clinically important differences in patient outcomes. I will need to include a range of methodological rigor to test this hypothesis. In addition, I added a citation from Barlow.
Your methods for analysis need to be more explicit. Please avoid the "qualitative meta-analysis" term as it feels like nails on a chalk-board. You call it a narrative synthesis in the text, which is a better term. See Petticrew's work on narrative syntheses.
I changed terminology throughout to "narrative synthesis"
In the abstract you mention using CONSORT (and the extension, which is appropriate) to assess the studies. However, CONSORT should not be used this way as it represents reporting guidelines and not risk of bias assessment. You mention the Cochrane RoB tool in your methods which is appropriate. Please edit your abstract for concurrence.
I changed the abstract to clarify we will use the Cochrane tool to assess the risk of bias of included studies. I will also use some components of the CENT checklist to collect data on completeness of reporting for n-of-1 specific design issues as listed in section 2.5 (data collection).
As I said above, I think this is an exciting concept, and look forward to reading both the protocol and the completed review. Best Wishes!
Reviewer 2 Report
In general, the topic is interesting and the proposed methods appropriate. I have however two major comments:
First, when I think about a n-of-1 trial, I have a type of study in mind that evaluates a specific treatment for a single subject (for instance by comparing the expected outcome level under a treatment condition with the outcome level when 'usual care' is given). At first view, it therefore looks strange to compare n-of-1 trials to usual care (the first one is an experimental design, the second is a treatment). I had to reread the protocol to understand that the authors consider an n-of-1 trial as an approach for selecting a treatment for an individual patient. The problem I have is that it still remains vague in the protocol what is understood by such a 'n-of-1 trial approach'. Without further specification, i am not convinced that both treatments being compared in the RCTs are clearly distinguished, it is not very clear to me how the results of the proposed review should be interpreted, and in my opinion the eligibility of specific studies remains unclear.
Second (but somewhat related to the first concern): the search terms that would be used by the authors all are more or less synonyms of n-of-1 trials. I think the authors are right that the types of RCTs they are interested in (RCTs that compare n-of-1 trial approaches for selecting a treatment on one side and usual care on the other side) are rare, but tons of n-of-1 trials are published (cfr. I worked on a review of quantitative n-of-1 trial meta-analyses in a specific field and found about 180 of these meta-analyses, so the number of primary studies using n-of-1 designs is a multiple thereof). I doubt whether the search terms are specific enough to find the few studies you are interested in (it is like looking for a needle in a haystack).
Author Response
(reviewer comments in italics, author response in bold)
Thank you for your thoughtful review and comments. All of your suggestions will be incorporated into the revision. See my point-by-point responses below.
In general, the topic is interesting and the proposed methods appropriate. I have however two major comments: First, when I think about a n-of-1 trial, I have a type of study in mind that evaluates a specific treatment for a single subject (for instance by comparing the expected outcome level under a treatment condition with the outcome level when 'usual care' is given). At first view, it therefore looks strange to compare n-of-1 trials to usual care (the first one is an experimental design, the second is a treatment). I had to reread the protocol to understand that the authors consider an n-of-1 trial as an approach for selecting a treatment for an individual patient. The problem I have is that it still remains vague in the protocol what is understood by such a 'n-of-1 trial approach'. Without further specification, i am not convinced that both treatments being compared in the RCTs are clearly distinguished, it is not very clear to me how the results of the proposed review should be interpreted, and in my opinion the eligibility of specific studies remains unclear.
Reviewer 1 raised similar concerns about the need to clarify that I am interested in evaluating the use of n-of-1 as a treatment selection approach in the context of therapeutic uncertainty in clinical practice. I rewrote the title, abstract, and introduction to explicitly describe the use of n-of-1 designs as a decision tool in clinical care as opposed to purely an experimental/research approach.
Second (but somewhat related to the first concern): the search terms that would be used by the authors all are more or less synonyms of n-of-1 trials. I think the authors are right that the types of RCTs they are interested in (RCTs that compare n-of-1 trial approaches for selecting a treatment on one side and usual care on the other side) are rare, but tons of n-of-1 trials are published (cfr. I worked on a review of quantitative n-of-1 trial meta-analyses in a specific field and found about 180 of these meta-analyses, so the number of primary studies using n-of-1 designs is a multiple thereof). I doubt whether the search terms are specific enough
The nomenclature issue is admittedly a legitimate problem, as n-of-1 trials have many synonyms and no MESH term exists yet. Given these limitations, the only feasible solution was to use the terms that we have seen others using to classify these papers. We believe we have exploited the current database structures as optimally as possible but we cannot exclude a form of cognitive bias in how these studies are labeled. We recognize that this will result in the need for an exhaustive review at the title/abstract level to exclude the irrelevant studies.
Round 2
Reviewer 2 Report
The description of the study now is much clearer. I think this can be an interesting study.
My concern about the large amount of hits that can be expected when looking for studies remains the same, but I do not think there is an easy solution for this.